# A SIMPLE SPARSE DENOISING LAYER FOR ROBUST DEEP LEARNING

## ABSTRACT

Deep models have achieved great success in many applications. However, vanilla deep models are not well-designed against the input perturbation. In this work, we take an initial step to design a simple robust layer as a lightweight plug-in for vanilla deep models. To achieve this goal, we first propose a fast sparse coding and dictionary learning algorithm for sparse coding problem with an exact $k$-sparse constraint or $l_0$ norm regularization. Our method comes with a closed-form approximation for the sparse coding phase by taking advantage of a novel structured dictionary. With this handy approximation, we propose a simple sparse denoising layer (SDL) as a lightweight robust plug-in. Extensive experiments on both classification and reinforcement learning tasks manifest the effectiveness of our methods.

## 1 INTRODUCTION

Deep neural networks have obtained a great success in many applications, including computer vision, reinforcement learning (RL) and natural language processing, etc. However, vanilla deep models are not robust to noise perturbations of the input. Even a small perturbation of input data would dramatically harm the prediction performance (Goodfellow et al., 2015).

To address this issue, there are three mainstreams of strategies: data argumentation based learning methods (Zheng et al., 2016; Ratner et al., 2017; Madry et al., 2018; Cubuk et al., 2020), loss functions/regularization techniques (Elsayed et al., 2018; Zhang et al., 2019), and importance weighting of network architecture against noisy input perturbation. Su et al. (2018) empirically investigated 18 deep classification models. Their studies found that model architecture is a more critical factor to robustness than the model size. Most recently, Guo et al. (2020) employed a neural architecture search (NAS) method to investigate the robust architectures. However, the NAS-based methods are still very computationally expensive. Furthermore, their resultant model cannot be easily adopted as a plug-in for other vanilla deep models. A handy robust plug-in for backbone models remains highly demanding.

In this work, we take an initial step to design a simple robust layer as a lightweight plug-in for the vanilla deep models. To achieve this goal, we first propose a novel fast sparse coding and dictionary learning algorithm. Our algorithm has a closed-form approximation for the sparse coding phase, which is cheap to compute compared with iterative methods in the literature. The closed-form update is handy for the situation that needs fast computation, especially in the deep learning. Based on this, we design a very simple sparse denoising layer for deep models. Our SDL is very flexible, and it enables an end-to-end training. Our SDL can be used as a lightweight plug-in for many modern architecture of deep models (e.g., ResNet and DenseDet for classification and deep PPO models for RL). Our contributions are summarized as follows:

- We propose simple sparse coding and dictionary learning algorithms for both k-sparse constrained sparse coding problem and $l_0$-norm regularized problem. Our algorithms have simple approximation form for the sparse coding phase.

- We introduce a simple sparse denoising layer (SDL) based on our handy update. Our SDL involves simple operations only, which is a fast plug-in layer for end-to-end training.

- Extensive experiments on both classification tasks and reinforcement learning tasks show the effectiveness of our SDL.

## 2 RELATED WORKS

**Sparse Coding and Dictionary Learning:** Sparse coding and dictionary learning are widely studied in computer vision and image processing. One related popular method is K-SVD (Elad & Aharon, 2006; Rubinstein et al., 2008), it jointly learns an over-complete dictionary and the sparse representations by minimizing a $l_0$-norm regularized reconstruction problem. Specifically, K-SVD alternatively iterates between the sparse coding phase and dictionary updating phase. The both steps are based on heuristic greedy methods. Despite its good performance, K-SVD is very computationally demanding. Moreover, as pointed out by Bao et al. (2013), both the sparse coding phase and dictonary updating of K-SVD use some greedy approaches that lack rigorous theoretical guarantee on its optimality and convergence. Bao et al. (2013) proposed to learn an orthogonal dictionary instead of the over-complete one. The idea is to concatenate the free parameters with predefined filters to form an orthogonal dictionary. This trick reduces the time complexity compared with K-SVD. However, their algorithm relies on the predefined filters. Furthermore, the alternative descent method heavily relies on SVD, which is not easy to extend to deep models.

In contrast, our method learns a structured over-complete dictionary, which has a simple form as a layer for deep learning. Recently, some works (Venkatakrishnan et al., 2013) employed deep neural networks to approximate alternating direction method of multipliers (ADMM) or other proximal algorithms for image denoising tasks. In (Wei et al., 2020), reinforcement learning is used to learn the hyperparameters of these deep iterative models. However, this kind of method itself needs to train a complex deep model. Thus, they are computationally expensive, which is too heavy or inflexible as a plug-in layer for backbone models in other tasks instead of image denoising tasks, e.g., reinforcement learning and multi-class classification, etc. An illustration of number of parameters of SDL, DnCNN (Zhang et al., 2017) and PnP (Wei et al., 2020) are shown in Table 1. SDL has much less parameters and simpler structure compared with DnCNN and PnP, and it can serve as a lightweight plug-in for other tasks, e.g., RL.

Table 1: Number of parameters of different models

|  | SDL | DnCNN (Zhang et al., 2017) | PnP (Wei et al., 2020) |
|---|---|---|---|
| #parameters | **630** | 667,008 | 296,640 |

**Robust Deep Learning:** In the literature of robust deep learning, several robust losses have been studied. To achieve better generalization ability, Elsayed et al. (2018) proposed a loss function to impose a large margin of any chosen layers of a deep network. Barron (2019) proposed a general loss with a shape parameter to cover several robust losses as special cases. For the problems with noisy input perturbation, several data argumentation-based algorithms and regularization techniques are proposed (Zheng et al., 2016; Ratner et al., 2017; Cubuk et al., 2020; Elsayed et al., 2018; Zhang et al., 2019). However, the network architecture remains less explored to address the robustness of the input perturbation. Guo et al. (2020) employed NAS methods to search the robust architectures. However, the searching-based method is very computationally expensive. The resultant architectures cannot be easily used as a plug-in for other popular networks. In contrast, our SDL is based on a closed-form of sparse coding, which can be used as a handy plug-in for many backbone models.

## 3 FAST SPARSE CODING AND DICTIONARY LEARNING

In this section, we present our fast sparse coding and dictionary learning algorithm for the $k$-sparse problem and the $l_0$-norm regularized problem in Section 3.1 and Section 3.2, respectively. Both algorithms belong to the alternative descent optimization framework.

### 3.1 k-SPARSE CODING

We first introduce the optimization problem for sparse coding with a $k$-sparse constraint. Mathematically, we aim at optimizing the following objective

$$\min_{Y,D} \|X - DY\|_F^2$$
$$\text{subject to} \ \ \|y_i\|_0 \le k, \forall i \in \{1, \cdots, N\} \tag{1}$$
$$\mu(D) \le \lambda$$
$$\|d_j\|_2 = 1, \forall j \in \{1, \cdots, M\},$$

where $\boldsymbol{D} \in \mathbb{R}^{d \times M}$ is the dictionary, and $\boldsymbol{d}_i$ denotes the $i^{th}$ column of matrix $\boldsymbol{D}$. $\boldsymbol{y}_i$ denotes the $i^{th}$ column of the matrix $\boldsymbol{Y} \in \mathbb{R}^{M \times N}$, and $\mu(\cdot)$ denotes the mutual coherence that is defined as

$$\mu(\boldsymbol{D}) = \max_{i \neq j} \frac{|\boldsymbol{d}_i^\top \boldsymbol{d}_j|}{\|\boldsymbol{d}_i\|_2 \|\boldsymbol{d}_j\|_2}. \tag{2}$$

The optimization problem (1) is discrete and non-convex, which is very difficult to optimize. To alleviate this problem, we employ a structured dictionary as

$$\boldsymbol{D} = \boldsymbol{R}^\top \boldsymbol{B}. \tag{3}$$

We require that $\boldsymbol{R}^\top \boldsymbol{R} = \boldsymbol{R}\boldsymbol{R}^\top = \boldsymbol{I}_d$ and $\boldsymbol{B}\boldsymbol{B}^\top = \boldsymbol{I}_d$, and each column vector of matrix $\boldsymbol{B}$ has a constant $l_2$-norm, i.e., $\|\boldsymbol{b}_i\|_2 = c$. The benefit of the structured dictionary is that it enables a fast update algorithm with a closed-form approximation for the sparse coding phase.

### 3.1.1 CONSTRUCTION OF STRUCTURED MATRIX $\boldsymbol{B}$

Now, we show how to design a structured matrix $\boldsymbol{B}$ that satisfies the requirements. First, we construct $\boldsymbol{B}$ by concatenating the real and imaginary parts of rows of a discrete Fourier matrix. The proof of the following theorems regarding the properties of $\boldsymbol{B}$ can be found in Appendix.

Without loss of generality, we assume that $d = 2m, M = 2n$. Let $\boldsymbol{F} \in \mathbb{C}^{n \times n}$ be an $n \times n$ discrete Fourier matrix. $\boldsymbol{F}_{k,j} = e^{\frac{2\pi i k j}{n}}$ is the $(k, j)^{th}$ entry of $\boldsymbol{F}$, where $\boldsymbol{i} = \sqrt{-1}$. Let $\Lambda = \{k_1, k_2, ..., k_m\} \subset \{1, ..., n-1\}$ be a subset of indexes.

The structured matrix $\boldsymbol{B}$ can be constructed as Eq.(4).

$$\boldsymbol{B} = \frac{1}{\sqrt{n}} \begin{bmatrix} \text{Re}\boldsymbol{F}_\Lambda & -\text{Im}\boldsymbol{F}_\Lambda \\ \text{Im}\boldsymbol{F}_\Lambda & \text{Re}\boldsymbol{F}_\Lambda \end{bmatrix} \in \mathbb{R}^{d \times N} \tag{4}$$

where Re and Im denote the real and imaginary parts of a complex number, and $\boldsymbol{F}_\Lambda$ in Eq. (5) is the matrix constructed by $m$ rows of $\boldsymbol{F}$

$$\boldsymbol{F}_\Lambda = \begin{bmatrix} e^{\frac{2\pi i k_1 1}{n}} & \cdots & e^{\frac{2\pi i k_1 n}{n}} \\ \vdots & \ddots & \vdots \\ e^{\frac{2\pi i k_m 1}{n}} & \cdots & e^{\frac{2\pi i k_m n}{n}} \end{bmatrix} \in \mathbb{C}^{m \times n}. \tag{5}$$

**Proposition 1.** *Suppose $d = 2m, M = 2n$. Construct matrix $\boldsymbol{B}$ as in Eq.(4). Then $\boldsymbol{B}\boldsymbol{B}^\top = \boldsymbol{I}_d$ and $\|\boldsymbol{b}_j\|_2 = \sqrt{\frac{m}{n}}, \forall j \in \{1, \cdots, M\}$.*

Theorem 1 shows that the structured construction $\boldsymbol{B}$ satisfies the orthogonal constraint and constant norm constraint. One thing remaining is how to construct $\boldsymbol{B}$ to achieve a small mutual coherence.

To achieve this goal, we can leverage the coordinate descent method in (Lyu, 2017) to construct the index set $\Lambda$. For a prime number $n$ such that $m$ divides $n-1$, i.e., $m|(n-1)$, we can employ a closed-form construction. Let $g$ denote a primitive root modulo $n$. We construct the index $\Lambda = \{k_1, k_2, ..., k_m\}$ as

$$\Lambda = \{g^0, g^{\frac{n-1}{m}}, g^{\frac{2(n-1)}{m}}, \cdots, g^{\frac{(m-1)(n-1)}{m}}\} \bmod n. \tag{6}$$

The resulted structured matrix $\boldsymbol{B}$ has a bounded mutual coherence, which is shown in Theorem 1.

**Theorem 1.** *Suppose $d = 2m, M = 2n$, and $n$ is a prime such that $m|(n-1)$. Construct matrix $\boldsymbol{B}$ as in Eq.(4) with index set $\Lambda$ as Eq.(6). Let mutual coherence $\mu(\boldsymbol{B}) := \max_{i \neq j} \frac{|\boldsymbol{b}_i^\top \boldsymbol{b}_j|}{\|\boldsymbol{b}_i\|_2 \|\boldsymbol{b}_j\|_2}$. Then $\mu(\boldsymbol{B}) \leq \frac{\sqrt{n}}{m}$.*

**Remark:** The bound of mutual coherence in Theorem 1 is non-trivial when $n < m^2$. For the case $n \geq m^2$, we can use the coordinate descent method in (Lyu, 2017) to minimize the mutual coherence.

Now, we show that the structured dictionary $\boldsymbol{D} = \boldsymbol{R}^\top \boldsymbol{B}$ satisfies the constant norm constraint and has a bounded mutual coherence. The results are summarized in Theorem 1.

**Corollary 1.** *Let $D = R^\top B$ with $R^\top R = RR^\top = I_d$. Construct matrix $B$ as in Eq.(4) with index set $\Lambda$ as Eq.(6). Then $\mu(D) = \mu(B) \leq \frac{\sqrt{n}}{m}$ and $\|d_j\|_2 = \|b_j\|_2 = \sqrt{\frac{m}{n}}, \forall j \in \{1, \cdots, M\}$.*

Corollary 1 shows that, for any orthogonal matrix $R$, each column vector of the structured dictionary $D$ has a constant $l_2$-norm. Moreover, it remains a constant mutual coherence $\mu(D) = \mu(B)$. Thus, given a fixed matrix $B$, we only need to learn matrix $R$ for the dictionary learning without undermining the low mutual coherence property.

### 3.1.2  Joint Optimization for Dictionary Learning and Sparse Coding

With the structured matrix $B$, we can jointly optimize $R$ and $Y$ for the optimization problem (7).

$$
\begin{aligned}
\min_{Y,R} &\ \|X - R^\top BY\|_F^2 \\
\text{subject to} &\ \|y_i\|_0 \leq k, \forall i \in \{1, \cdots, N\} \\
&\ R^\top R = RR^\top = I_d
\end{aligned}
\tag{7}
$$

This problem can be solved by the alternative descent method. For a fixed $R$, we show the sparse representation $Y$ has a closed-form approximation thanks to the structured dictionary. For the fixed sparse codes $Y$, dictionary parameter $R$ has a closed-form solution.

**Fix $R$, optimize $Y$:** Since the constraints of $Y$ is column separable, i.e., $\|y_i\|_0 \leq k$, and the objective (8) is also decomposable,

$$
\|X - R^\top BY\|_F^2 = \sum_{i=1}^N \|x_i - R^\top By_i\|_F^2.
\tag{8}
$$

It is sufficient to optimize the sparse code $y_i \in \mathbb{R}^M$ for each point $x_i \in \mathbb{R}^d$ separately.

Without loss of generality, for any input $x \in \mathbb{R}^d$, we aim at finding the optimal sparse code $y \in \mathbb{R}^M$ such that $\|y\|_0 \leq k$. Since $R^\top R = RR^\top = I_d$ and $BB^\top = I_d$, we have

$$
\begin{aligned}
\|x - R^\top By\|_2^2 &= \|Rx - RR^\top By\|_2^2 \\
&= \|Rx - By\|_2^2 \\
&= \|BB^\top Rx - By\|_2^2 \\
&= \|B(B^\top Rx - y)\|_2^2 \\
&= \|B(h - y)\|_2^2.
\end{aligned}
\tag{9}
$$

where $h = B^\top Rx$ is a dense code. **Case 1:** When $m = n$ (the columns of $B$ are orthogonal), we can rewrite Eq.(9) into a summation form as

$$
\|B(h - y)\|_2^2 = \sum_{j=1}^M (h_j - y_j)^2 \|b_j\|_2^2.
\tag{10}
$$

**Case 2:** When $m < n$, we have an error-bounded approximation using R.H.S. in Eq.(10). Let $z = h - y$, we have

$$
\left| \|Bz\|_2^2 - \sum_{j=1}^M z_j^2 \|b_j\|_2^2 \right| = \left| \sum_{i=1}^M \sum_{j=1, j\neq i}^M z_i z_j b_i^\top b_j \right|
\tag{11}
$$

$$
\leq \sum_{i=1}^M \sum_{j=1, j\neq i}^M |z_i z_j| \|b_i\|_2 \|b_j\|_2 \mu(B)
\tag{12}
$$

$$
= \sum_{i=1}^M \sum_{j=1, j\neq i}^M |z_i z_j| \cdot \frac{m}{n} \cdot \mu(B)
\tag{13}
$$

It is worth to note that the error bound is small when the mutual coherence $\mu(\boldsymbol{B})$ is small. When we employ the structural matrix in Theorem 1. It follows that

$$\left|\|\boldsymbol{Bz}\|_2^2 - \sum_{j=1}^M z_j^2 \|\boldsymbol{b}_j\|_2^2\right| \leq \sum_{i=1}^M \sum_{j=1, j\neq i}^M |z_i z_j| \cdot \frac{m}{n} \cdot \min(\frac{\sqrt{n}}{m}, 1) \tag{14}$$

$$= C \sum_{i=1}^M \sum_{j=1, j\neq i}^M |z_i z_j| \tag{15}$$

$$= C \sum_{i=1}^M \sum_{j=1, j\neq i}^M |h_i - y_i||h_j - y_j| \tag{16}$$

where $C = \min(\frac{1}{\sqrt{n}}, \frac{m}{n})$. In Eq.(14), we use $\mu(\boldsymbol{B}) \leq \frac{\sqrt{n}}{m}$ from Theorem 1.

Considering the sparse constraint $\|\boldsymbol{y}\|_0 \leq k$, the error bound is minimized when all the non-zero term $y_j = h_j$ to get $|y_j - h_j| = 0$. Let $S$ denote the set of index of non-zero element $y_j$ of $\boldsymbol{y}$. Now the problem is to find the index set $S$ to minimize

$$\sum_{i=1}^M \sum_{j=1, j\neq i}^M |h_i - y_i||h_j - y_j| = \sum_{i\in S^c} \sum_{j\in S^c, j\neq i} |h_i||h_j| \tag{17}$$

where $S^c$ denotes the complement set of $S$. We can see that Eq.(17) is minimized when $S$ consists of the index of the k largest (in absolute value) elements of $\boldsymbol{h}$.

Now, we consider $\sum_{j=1}^M z_j^2 \|\boldsymbol{b}_j\|_2^2$. Note that $\|\boldsymbol{b}_j\|_2^2 = \frac{m}{n}$, it follows that

$$\sum_{j=1}^M z_j^2 \|\boldsymbol{b}_j\|_2^2 = \frac{m}{n} \sum_{j=1}^M (h_j - y_j)^2. \tag{18}$$

Because each term $(h_j - y_j)^2 \geq 0$ is minimized when $y_j = h_j$, we know that Eq.(18) under sparse constraints is minimized when all the non-zero term setting as $y_j = h_j$. Otherwise we can set a non-zero term $y_j$ to $y_j = h_j$ to further reduce term $(h_j - y_j)^2$ to zero.

Now, the problem is to find the index set of the non-zero term to minimize Eq.(19).

$$\sum_{j=1}^M (h_j - y_j)^2 = \sum_{j=1}^M h_j^2 - \sum_{i\in S, |S|\leq k} h_i^2 \tag{19}$$

where $S := \{j|y_j \neq 0\}$. We can see that Eq.(19) is minimized when $S$ consists of the index of the k largest (in absolute value ) elements of $\boldsymbol{h}$.

**Remark:** Both the approximation $\sum_{j=1}^M z_j^2 \|\boldsymbol{b}_j\|_2^2$ and the error bound is minimized by the same solution.

**Fix $\boldsymbol{Y}$, Optimize $\boldsymbol{R}$** : For a fixed $\boldsymbol{Y}$, we know that

$$\|\boldsymbol{X} - \boldsymbol{R}^\top \boldsymbol{BY}\|_F^2 = \|\boldsymbol{X}\|_F^2 + \|\boldsymbol{BY}\|_F^2 - 2\mathrm{tr}(\boldsymbol{R}^\top \boldsymbol{BY}\boldsymbol{X}^\top) \tag{20}$$

This is the nearest orthogonal matrix problem, which has a closed-form solution as shown in (Schönemann, 1966; Gong et al., 2012). Let $\boldsymbol{BYX}^\top = \boldsymbol{U}\boldsymbol{\Gamma}\boldsymbol{V}^\top$ obtained by singular value decomposition (SVD), where $\boldsymbol{U}, \boldsymbol{V}$ are orthogonal matrix. Then, Eq.(20) is minimized by

$$\boldsymbol{R} = \boldsymbol{U}\boldsymbol{V}^\top \tag{21}$$

## 3.2 $l_0$-NORM REGULARIZATION

We employ the structured dictionary $\boldsymbol{D} = \boldsymbol{R}^\top \boldsymbol{B}$ same as in Section 3.1. The optimization problem with $l_0$-norm regularization is defined as

$$\min_{\boldsymbol{Y},\boldsymbol{R}} \|\boldsymbol{X} - \boldsymbol{R}^\top \boldsymbol{BY}\|_F^2 + \lambda \|\boldsymbol{Y}\|_0$$

$$\text{subject to } \boldsymbol{R}^\top \boldsymbol{R} = \boldsymbol{RR}^\top = \boldsymbol{I}_d \tag{22}$$

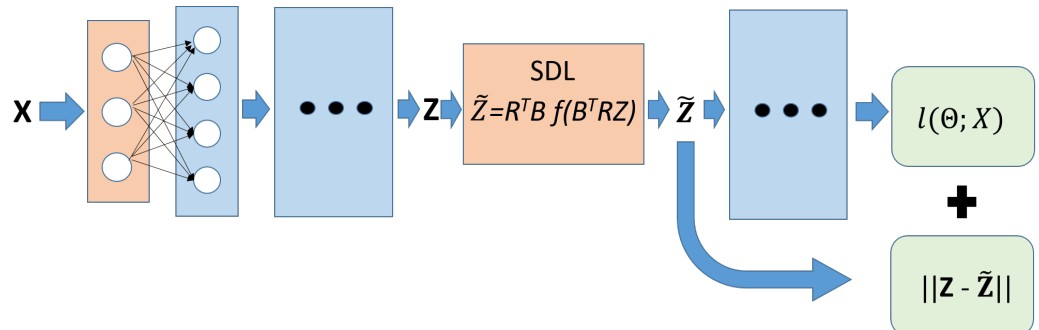

Figure 1: Illustration of the SDL Plug-in

This problem can be solved by the alternative descent method. For a fixed $\boldsymbol{R}$, we show $\boldsymbol{Y}$ has a closed-form approximation thanks to the structured dictionary. For fixed the sparse codes $\boldsymbol{Y}$, dictionary parameter $\boldsymbol{R}$ also has a closed-form solution.

**Fix $\boldsymbol{R}$, optimize $\boldsymbol{Y}$:** Since the objective can be rewritten as Eq.(23)

$$\|\boldsymbol{X} - \boldsymbol{R}^\top \boldsymbol{B}\boldsymbol{Y}\|_F^2 + \lambda\|\boldsymbol{Y}\|_0 = \sum_{i=1}^{N} \|\boldsymbol{x}_i - \boldsymbol{R}^\top \boldsymbol{B}\boldsymbol{y}_i\|_F^2 + \lambda\|\boldsymbol{y}_i\|_0. \tag{23}$$

It is sufficient to optimize $\boldsymbol{Y}_i$ for each point $\boldsymbol{X}_i$ separately. Without loss of generality, for any input $\boldsymbol{x} \in \mathbb{R}^d$, we aim at finding an optimal sparse code $\boldsymbol{y} \in \mathbb{R}^M$. Since $\boldsymbol{R}^\top \boldsymbol{R} = \boldsymbol{R}\boldsymbol{R}^\top = \boldsymbol{I}_d$ and $\boldsymbol{B}\boldsymbol{B}^\top = \boldsymbol{I}_d$ , when $m = n$, following the derivation in Section 3.1.2, we have

$$\|\boldsymbol{x} - \boldsymbol{R}^\top \boldsymbol{B}\boldsymbol{y}\|_F^2 + \lambda\|\boldsymbol{y}\|_0 = \|\boldsymbol{B}(\boldsymbol{h} - \boldsymbol{y})\|_F^2 + \lambda\|\boldsymbol{y}\|_0, \tag{24}$$

where $\boldsymbol{h} = \boldsymbol{B}^\top \boldsymbol{R}\boldsymbol{x}$ is a dense code. Note that $\|\boldsymbol{b}_j\|_2^2 = \frac{m}{n}$, together with Eq.(24), it follows that

$$\|\boldsymbol{B}(\boldsymbol{h} - \boldsymbol{y})\|_F^2 + \lambda\|\boldsymbol{y}\|_0 = \frac{m}{n}\left(\sum_{j=1}^{M}(h_j - y_j)^2 + \frac{n\lambda}{m}\mathbf{1}[y_j \neq 0]\right). \tag{25}$$

where $\mathbf{1}[\cdot]$ is an indicator function which is 1 if its argument is true and 0 otherwise.

This problem is separable for each variable $y_j$, and each term is minimized by setting

$$y_j = \begin{cases} h_j & \text{if } h_j^2 \geq \frac{n\lambda}{m} \\ 0 & \text{otherwise} \end{cases}. \tag{26}$$

**Fix $\boldsymbol{Y}$, update $\boldsymbol{R}$:** For a fixed $\boldsymbol{Y}$, minimizing the objective leads to the same nearest orthogonal matrix problem as shown in Section 3.1.2. Let $\boldsymbol{B}\boldsymbol{Y}\boldsymbol{X}^\top = \boldsymbol{U}\Gamma\boldsymbol{V}^\top$ obtained by SVD, where $\boldsymbol{U}, \boldsymbol{V}$ are orthogonal matrix. Then, the reconstruction problem is minimized by $\boldsymbol{R} = \boldsymbol{U}\boldsymbol{V}^\top$.

**Remark:** Problems with other separable regularization terms can be solved in a similar way. The key difference is how to achieve sparse codes $\boldsymbol{y}$. For example, for $l_1$-norm regularized problems, $\boldsymbol{y}$ can be obtained by a soft thresholding function, i.e., $\boldsymbol{y} = \text{sign}(\boldsymbol{y}) \odot \max\left(0, |\boldsymbol{y}| - n\lambda/(2m)\right)$.

## 4 Sparse Denoising Layer

One benefit of our fast sparse coding algorithm is that it enables a simple closed-form reconstruction, which can be used as a plug-in layer for deep neural networks. Specifically, given an orthogonal matrix $\boldsymbol{R}$ and input vector $\boldsymbol{x}$, the optimal reconstruction of our method can be expressed as

$$\widetilde{\boldsymbol{x}} = \boldsymbol{R}^\top \boldsymbol{B}f(\boldsymbol{B}^\top \boldsymbol{R}\boldsymbol{x}), \tag{27}$$

where $f(\cdot)$ is a non-linear mapping function. For the k-sparse constrained problem, $f(\cdot)$ is a k-max pooling function (w.r.t the absolute value) as Eq.(28)

$$f(h_j) = \begin{cases} h_j & \text{if } |h_j| \text{ is one of the k-highest values of } |\boldsymbol{h}| \in \mathbb{R}^M \\ 0 & \text{otherwise} \end{cases}. \tag{28}$$

For the $l_0$-norm regularization problem, $f(\cdot)$ is a hard thresholding function as Eq.(29)

$$f(h_j) = \begin{cases} h_j & \text{if } |h_j| \geq \sqrt{\frac{n\lambda}{m}} \\ 0 & \text{otherwise} \end{cases}. \tag{29}$$

For the $l_1$-norm regularization problem, $f(\cdot)$ is a soft thresholding function as Eq.(30)

$$f(h_j) = \text{sign}(h_j) \times \max\left(0, |h_j| - \frac{n\lambda}{2m}\right), \tag{30}$$

where $\text{sign}(\cdot)$ denotes the Sign function.

The reconstruction in Eq.(27) can be used as a simple plug-in layer for deep networks, we named it as sparse denoising layer (SDL). It is worth noting that only the orthogonal matrix $\boldsymbol{R}$ is needed to learn. The structured matrix $\boldsymbol{B}$ is constructed as in Section 3.1.1 and fixed.

The orthogonal matrix $\boldsymbol{R}$ can be parameterized by exponential mapping or Cayley mapping (Helfrich et al., 2018) of a skew-symmetric matrix. In this work, we employ the Cayley mapping to enable gradient update using deep learning tools. Specifically, the orthogonal matrix $\boldsymbol{R}$ can be obtained by the Cayley mapping of a skew-symmetric matrix as

$$\boldsymbol{R} = (\boldsymbol{I} + \boldsymbol{W})(\boldsymbol{I} - \boldsymbol{W})^{-1}, \tag{31}$$

where $\boldsymbol{W}$ is a skew-symmetric matrix, i.e., $\boldsymbol{W} = -\boldsymbol{W}^\top \in \mathbb{R}^{d \times d}$. For a skew-symmetric matrix $\boldsymbol{W}$, only the upper triangular matrix (without main diagonal) are free parameters. Thus, the number of free parameters of SDL is $d(d-1)/2$, which is much smaller compared with the number of parameters of backbone deep networks.

For training a network with a SDL, we add a reconstruction loss term as a regularization. The optimization problem is defined as

$$\min_{\boldsymbol{W} \in \mathbb{R}^{d \times d}, \boldsymbol{\Theta}} \ell(\boldsymbol{X}; \boldsymbol{W}, \boldsymbol{\Theta}) + \beta \|\widetilde{\boldsymbol{Z}} - \boldsymbol{Z}\|_F^2, \tag{32}$$

where $\boldsymbol{W}$ is a skew-symmetric matrix parameter of SDL. $\boldsymbol{\Theta}$ is the parameter of the backbone network. $\widetilde{\boldsymbol{Z}}$ is the reconstruction of the latent representation $\boldsymbol{Z}$ via SDL (Eq.(27)). An illustration of the SDL plug-in is shown in Figure 1. When SDL is used as the first layer, then $\boldsymbol{Z} = \boldsymbol{X}$, and $\widetilde{\boldsymbol{Z}} = \widetilde{\boldsymbol{X}}$. In this case, $\widetilde{\boldsymbol{Z}}$ is the reconstruction of the input data $\boldsymbol{X}$.

It is worth noting that the shape of the input and output of SDL are same. Thus, SDL can be used as plug-in for any backbone models without changing the input/output shape of different layers in the backbone network. With the simple SDL plug-in, backbone models can be trained from scratches.

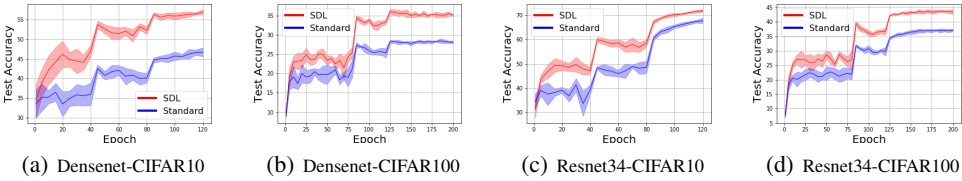

(a) Densenet-CIFAR10    (b) Densenet-CIFAR100    (c) Resnet34-CIFAR10    (d) Resnet34-CIFAR100

Figure 2: Mean test accuracy $\pm$ std over 5 independent runs on CIFAR10/CIFAR100 dataset under FGSM adversarial attack for Densenet and Resnet with or without SDL

## 5 EXPERIMENTS

We evaluate the performance of our SDL on both classification tasks and RL tasks. For classification, we employ both DenseNet-100 (Huang et al., 2017) and ResNet-34 (He et al., 2016) as backbone. For RL tasks, we employ deep PPO models[1] (Schulman et al., 2017) as backbone. For all the tasks, we test the performance of backbone models with and without our SDL when adding Gaussian noise or Laplace noise. In all the experiments, we plug SDL as the first layer of deep models. We set the

---

[1]https://github.com/ikostrikov/pytorch-a2c-ppo-acktr-gail/

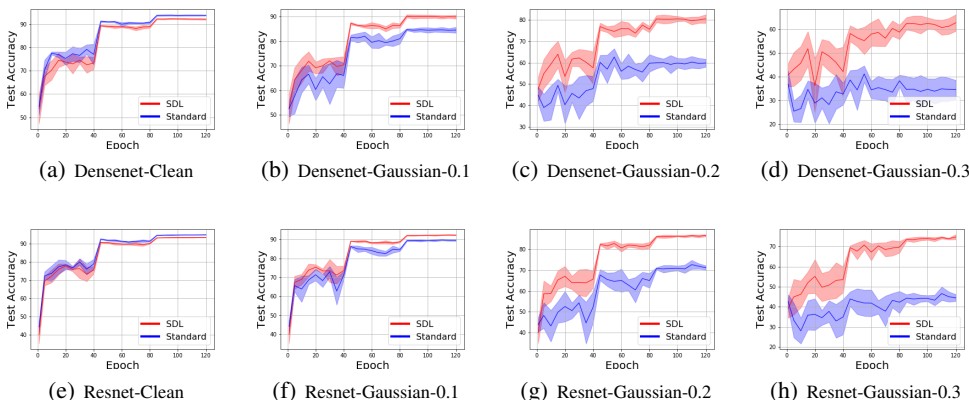

Figure 3: Mean test accuracy $\pm$ std over 5 independent runs on CIFAR10 dataset with Gaussian noise for Densenet and Resnet with or without SDL

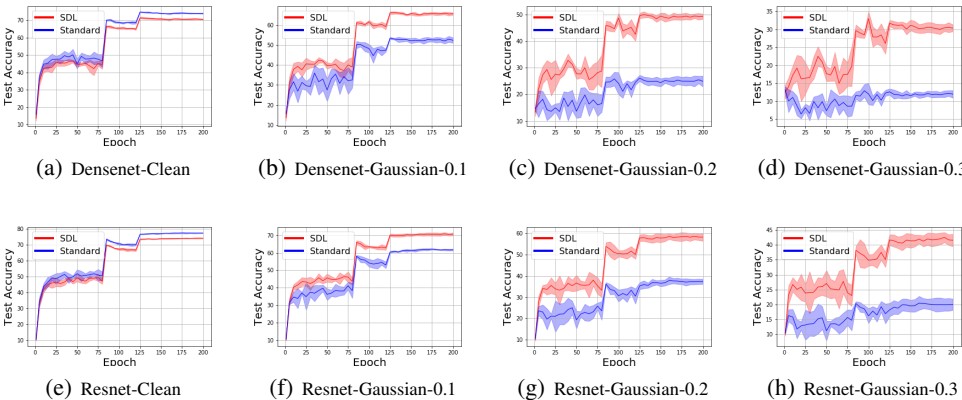

Figure 4: Mean test accuracy $\pm$ std over 5 independent runs on CIFAR100 dataset with Gaussian noise for Densenet and Resnet with or without SDL

standard deviation of input noise as $\{0, 0.1, 0.2, 0.3\}$, respectively. (The input noise is added after input normalization) . We keep all the hyperparameters of the backbone models same, the only difference is whether plugging SDL. The parameter $\beta$ for the reconstruction loss is fixed as $\beta = 100$ in all the experiments.

**Classification Tasks:** We test SDL on CIFAR10 and CIFAR100 dataset. We construct the structured matrix $\boldsymbol{B} \in \mathbb{R}^{12 \times 14}$ by Eq.(4). In this setting, the orthogonal matrix $\boldsymbol{R}$ corresponds to the convolution parameter of $Conv2d(\cdot)$ with *kernelsize*=$2 \times 2$. We set the sparse parameter of our $k$-sparse SDL as $k = 3$ in all the classification experiments. The average test accuracy over five independent runs on CIFAR10 and CIFAR100 with Gaussian noise are shown in Fig. 3 and Fig. 4, respectively. We can observe that models with SDL obtain a similar performance compared with vanilla model on the clean input case. With an increasing variance of input noise, models with SDL outperform the vanilla models more and more significantly. The experimental results with Laplace noise are presented in Fig. 9 and Fig. 10 in the supplementary. The results on Laplace noise cases show the similar trends with Gaussian noise cases. We further test the performance of SDL under the fast gradient sign method (FGSM) attack (Goodfellow et al., 2015). The perturbation parameter epsilon is set to 8/256. Experimental results are shown in Fig. 2. We can observe that adding SDL plug-in can improve the adversarial robustness of the backbone models. More experiments on tiny-Imagenet dataset can be found in Appendix G.

**RL Tasks:** We test deep PPO model with SDL on Atari games: *KungFuMaster*, *Tennis* and *Seaquest*. The deep PPO model concatenates four frames as the input state. The size of input

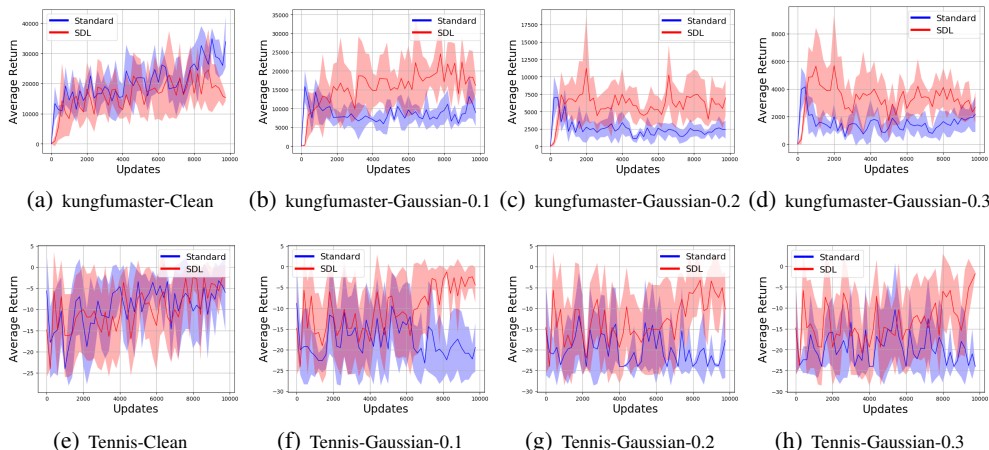

(a) kungfumaster-Clean  (b) kungfumaster-Gaussian-0.1  (c) kungfumaster-Gaussian-0.2  (d) kungfumaster-Gaussian-0.3

(e) Tennis-Clean  (f) Tennis-Gaussian-0.1  (g) Tennis-Gaussian-0.2  (h) Tennis-Gaussian-0.3

Figure 5: Average Return $\pm$ std over 5 independent runs on *KungfuMaster* and *Tennis* game with Gaussian noise with or without SDL

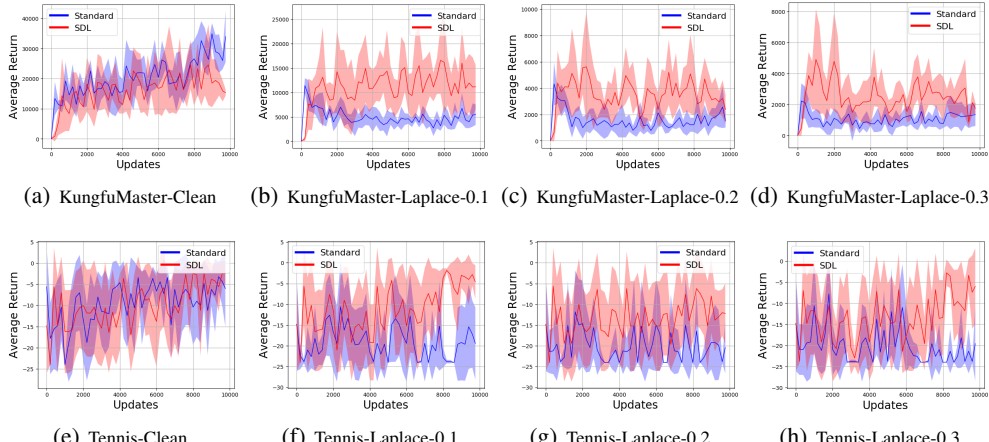

(a) KungfuMaster-Clean  (b) KungfuMaster-Laplace-0.1  (c) KungfuMaster-Laplace-0.2  (d) KungfuMaster-Laplace-0.3

(e) Tennis-Clean  (f) Tennis-Laplace-0.1  (g) Tennis-Laplace-0.2  (h) Tennis-Laplace-0.3

Figure 6: Average Return $\pm$ std over 5 independent runs on *KungfuMaster* and *Tennis* game with Laplace noise with or without SDL

state is $84 \times 84 \times 4$. We construct the structured matrix $\boldsymbol{B} \in \mathbb{R}^{36 \times 38}$ by Eq.(4). In this setting, the orthogonal matrix $\boldsymbol{R}$ corresponds to the convolution parameter of $Conv2d(\cdot)$ with *kernelsize*=$3 \times 3$. The number of free parameters is 630. We set the sparse parameter of our $k$-sparse SDL as $k = 4$ in all the RL experiments.

The return of one episode is the sum of rewards over all steps during the whole episode. We present the average return over five independent runs on *KungFuMaster* and *Tennis* game with Gaussian noise and Laplace noise in Fig. 5 and Fig. 6, respectively. Results on *Seaquest* game are shown in Fig. 16 in the supplement due to the space limitation. We can see that models with SDL achieve a competitive average return on the clean cases. Moreover, models with SDL obtain higher average return than vanilla models when the input state is perturbed with noise.

## 6 CONCLUSION

We proposed fast sparse coding algorithms for both $k$-sparse problem and $l_0$-norm regularization problems. Our algorithms have a simple closed-form update. We proposed a sparse denoising layer as a lightweight plug-in for backbone models against noisy input perturbation based on this handy closed-form. Experiments on both ResNet/DenseNet classification model and deep PPO RL model showed the effeteness of our SDL against noisy input perturbation and adversarial perturbation.

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

## A    PROOF OF PROPOSITION 1

*Proof.* Let $c_i \in \mathbb{C}^{1 \times n}$ be the $i^{th}$ row of matrix $F_\Lambda \in \mathbb{C}^{m \times n}$ in Eq.(5). Let $v_i \in \mathbb{R}^{1 \times 2n}$ be the $i^{th}$ row of matrix $B \in \mathbb{R}^{2m \times 2n}$ in Eq.(4). For $1 \leq i, j \leq m, i \neq j$, we know that

$$v_i v_{i+m}^\top = 0, \tag{33}$$

$$v_{i+m} v_{j+m}^\top = v_i v_j^\top = \mathrm{Re}(c_i c_j^*), \tag{34}$$

$$v_{i+m} v_j^\top = -v_i v_{j+m}^\top = \mathrm{Im}(c_i c_j^*), \tag{35}$$

where $*$ denotes the complex conjugate, $\mathrm{Re}(\cdot)$ and $\mathrm{Im}(\cdot)$ denote the real and imaginary parts of the input complex number.

For a discrete Fourier matrix $F$, we know that

$$c_i c_j^* = \frac{1}{n} \sum_{k=0}^{n-1} e^{\frac{2\pi(i-j)ki}{n}} = \begin{cases} 1, & \text{if } i = j \\ 0, & \text{otherwise} \end{cases} \tag{36}$$

When $i \neq j$, from Eq.(36), we know $c_i c_j^* = 0$. Thus, we have

$$v_{i+m} v_{j+m}^\top = v_i v_j^\top = \mathrm{Re}(c_i c_j^*) = 0, \tag{37}$$

$$v_{i+m} v_j^\top = -v_i v_{j+m}^\top = \mathrm{Im}(c_i c_j^*) = 0, \tag{38}$$

When $i = j$, we know that $v_{i+m} v_{i+m}^\top = v_i v_i^\top = c_i c_i^* = 1$.

Put two cases together, also note that $d = 2m$, we have $BB^\top = I_d$.

The $l_2$-norm of the column vector of $B$ is given as

$$\|b_j\|_2^2 = \frac{1}{n} \sum_{i=1}^{m} \left( \sin^2 \frac{2\pi k_i j}{n} + \cos^2 \frac{2\pi k_i j}{n} \right) = \frac{m}{n} \tag{39}$$

Thus, we have $\|b_j\|_2 = \sqrt{\frac{m}{n}}$ for $j \in \{1, \cdots, M\}$

$\square$

## B    PROOF OF THEOREM 1

*Proof.* Let $c_i \in \mathbb{C}^{m \times 1}$ be the $i^{th}$ column of matrix $F_\Lambda \in \mathbb{C}^{m \times n}$ in Eq.(5). Let $b_i \in \mathbb{R}^{2m \times 1}$ be the $i^{th}$ column of matrix $B \in \mathbb{R}^{2m \times 2n}$ in Eq.(4). For $1 \leq i, j \leq n, i \neq j$, we know that

$$b_i^\top b_{i+n} = 0, \tag{40}$$

$$b_{i+n}^\top b_{j+n} = b_i^\top b_j = \mathrm{Re}(c_i^* c_j), \tag{41}$$

$$b_{i+n}^\top b_j = -b_i^\top b_{j+n} = \mathrm{Im}(c_i^* c_j), \tag{42}$$

where $*$ denotes the complex conjugate, $\mathrm{Re}(\cdot)$ and $\mathrm{Im}(\cdot)$ denote the real and imaginary parts of the input complex number.

It follows that

$$\mu(B) \leq= \max_{1 \leq k, r \leq 2n, k \neq r} |b_k^\top b_r| \leq \max_{1 \leq i, j \leq n, i \neq j} |c_i^* c_j| = \mu(F_\Lambda) \tag{43}$$

From the definition of $F_\Lambda$ in Eq.(5), we know that

$$\mu(F_\Lambda) = \max_{1 \leq i, j \leq n, i \neq j} |c_i^* c_j| = \max_{1 \leq i, j \leq n, i \neq j} \frac{1}{m} \left| \sum_{z \in \Lambda} e^{\frac{2\pi i z(j-i)}{n}} \right| \tag{44}$$

$$= \max_{1 \leq k \leq n-1} \frac{1}{m} \left| \sum_{z \in \Lambda} e^{\frac{2\pi i z k}{n}} \right| \tag{45}$$

Because $\Lambda = \{g^0, g^{\frac{n-1}{m}}, g^{\frac{2(n-1)}{m}}, \cdots, g^{\frac{(m-1)(n-1)}{m}}\}$ mod $n$ , we know that $\Lambda$ is a subgroup of the multiplicative group $\{g^0, g^1, \cdots, g^{n-2}\}$ mod $n$. From Bourgain et al. (2006), we know that

$$\max_{1 \leq k \leq n-1} \left| \sum_{z \in \Lambda} e^{\frac{2\pi i z k}{n}} \right| \leq \sqrt{n} \tag{46}$$

Finally, we know that

$$\mu(\boldsymbol{B}) \leq \mu(\boldsymbol{F}_\Lambda) \leq \frac{\sqrt{n}}{m}. \tag{47}$$

$\square$

## C   PROOF OF COROLLARY 1

*Proof.* Since $\boldsymbol{R}^\top \boldsymbol{R} = \boldsymbol{R}\boldsymbol{R}^\top = \boldsymbol{I}_d$ and $\boldsymbol{D} = \boldsymbol{R}^\top \boldsymbol{B}$, we know that $\|\boldsymbol{d}_j\|_2 = \|\boldsymbol{b}_j\|_2$. From Theorem 1, we know that $\|\boldsymbol{b}_j\|_2 = \sqrt{\frac{m}{n}}$, $\forall j \in \{1, \cdots, M\}$. It follows that $\|\boldsymbol{d}_j\|_2 = \|\boldsymbol{b}_j\|_2 = \sqrt{\frac{m}{n}}$ for $\forall j \in \{1, \cdots, M\}$.

From the definition of mutual coherence $\mu(\cdot)$, we know it is rotation invariant. Since $\boldsymbol{D} = \boldsymbol{R}^\top \boldsymbol{B}$ with $\boldsymbol{R}^\top \boldsymbol{R} = \boldsymbol{R}\boldsymbol{R}^\top = \boldsymbol{I}_d$, we know $\mu(\boldsymbol{D}) = \mu(\boldsymbol{B})$. From Theorem 1, we have $\mu(\boldsymbol{B}) \leq \frac{\sqrt{n}}{m}$. Thus, we obtain $\mu(\boldsymbol{D}) = \mu(\boldsymbol{B}) \leq \frac{\sqrt{n}}{m}$.

$\square$

## D   EMPIRICAL CONVERGENCE OF OBJECTIVE FUNCTIONS

We test our fast dictionary learning algorithms on Lena with image patches (size $12 \times 12$). We present the empirical convergence result of our fast algorithms in Figure 7. It shows that the objective tends to converge less than fifty iterations.

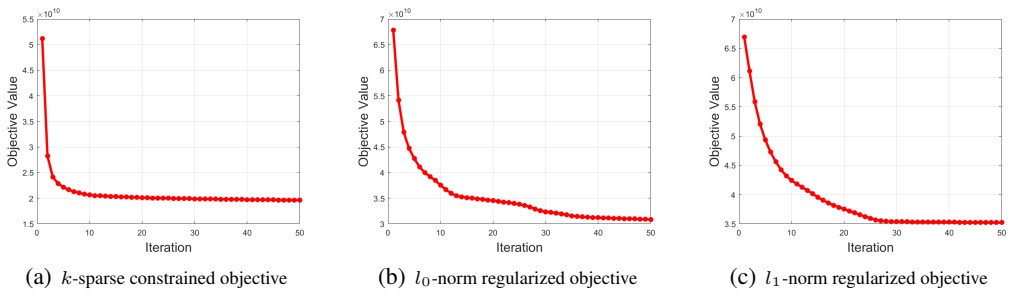

(a) $k$-sparse constrained objective       (b) $l_0$-norm regularized objective       (c) $l_1$-norm regularized objective

Figure 7:  Decreasing of the objective functions

## E  DEMO OF DENOISED IMAGES

We show the denoised results of our fast sparse coding algorithm on some widely used testing images. The input images are perturbed by Gaussian noise with std $\sigma = 100$. The denoised results are presented in Figure 8. It shows that our algorithms can reduce the influence of the noisy perturbation of the images.

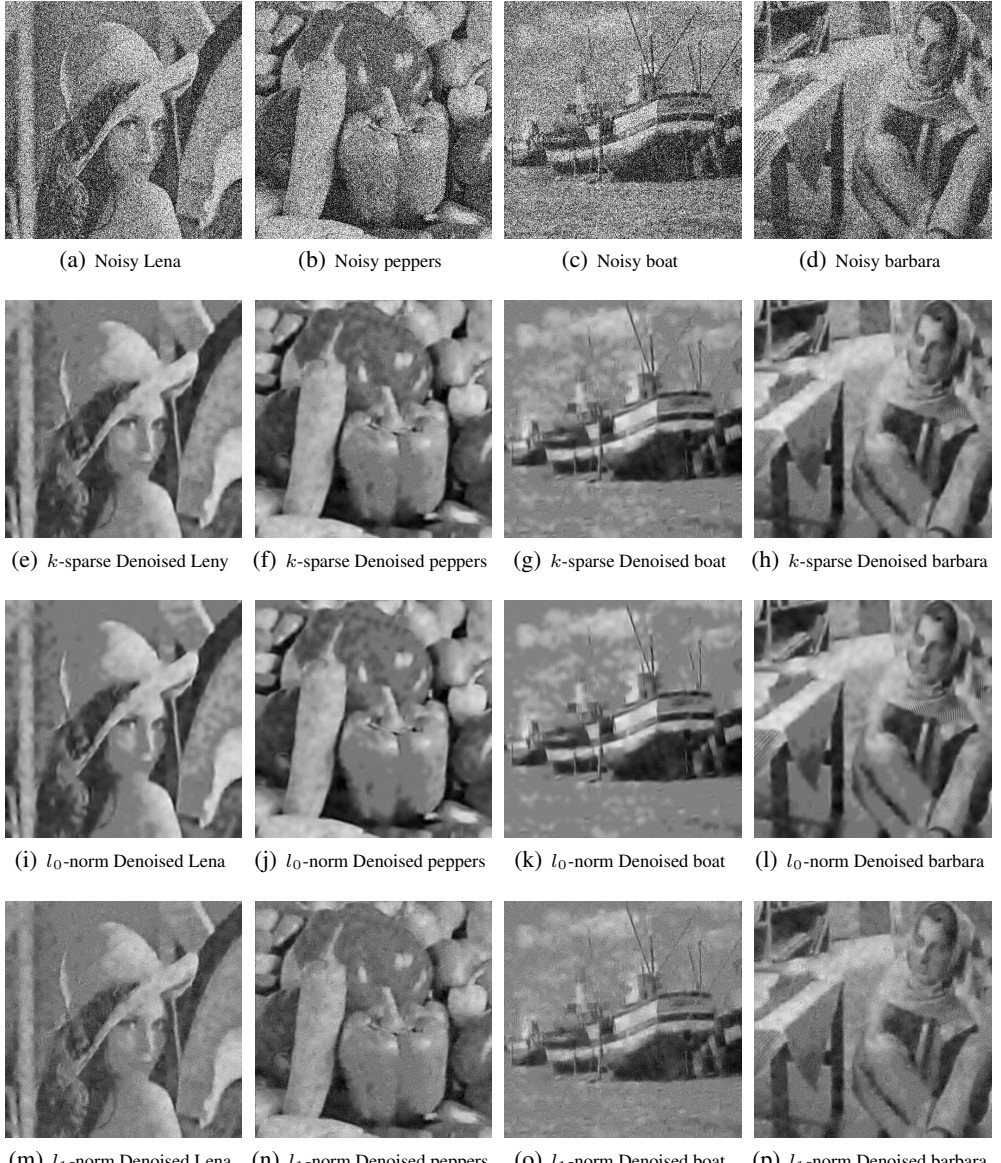

(a) Noisy Lena  (b) Noisy peppers  (c) Noisy boat  (d) Noisy barbara

(e) $k$-sparse Denoised Leny  (f) $k$-sparse Denoised peppers  (g) $k$-sparse Denoised boat  (h) $k$-sparse Denoised barbara

(i) $l_0$-norm Denoised Lena  (j) $l_0$-norm Denoised peppers  (k) $l_0$-norm Denoised boat  (l) $l_0$-norm Denoised barbara

(m) $l_1$-norm Denoised Lena  (n) $l_1$-norm Denoised peppers  (o) $l_1$-norm Denoised boat  (p) $l_1$-norm Denoised barbara

Figure 8: Demo of denoised results of our fast sparse coding algorithm

## F EXPERIMENTAL RESULTS ON CLASSIFICATION WITH LAPLACE NOISE

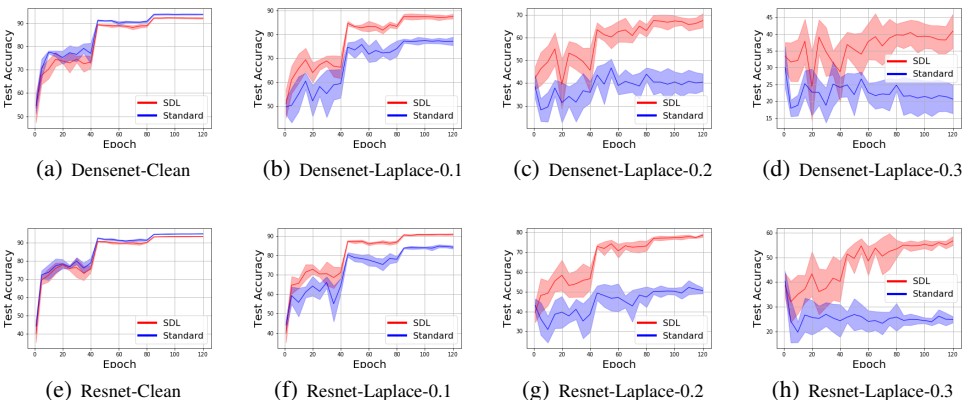

Figure 9: Mean test accuracy ± std over 5 independent runs on CIFAR10 dataset with Laplace noise for Densenet and Resnet with or without SDL

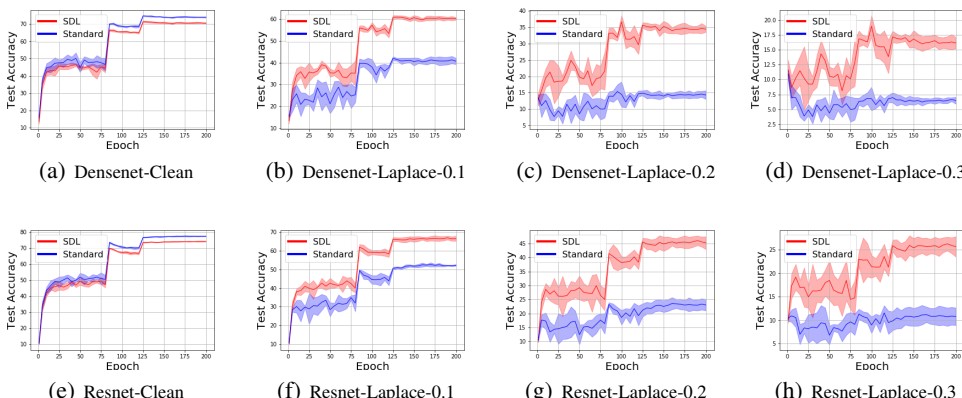

Figure 10: Mean test accuracy ± std over 5 independent runs on CIFAR100 dataset with Laplace noise for Densenet and Resnet with or without SDL

# G    EXPERIMENTAL RESULTS ON TINY-IMAGENET DATASET

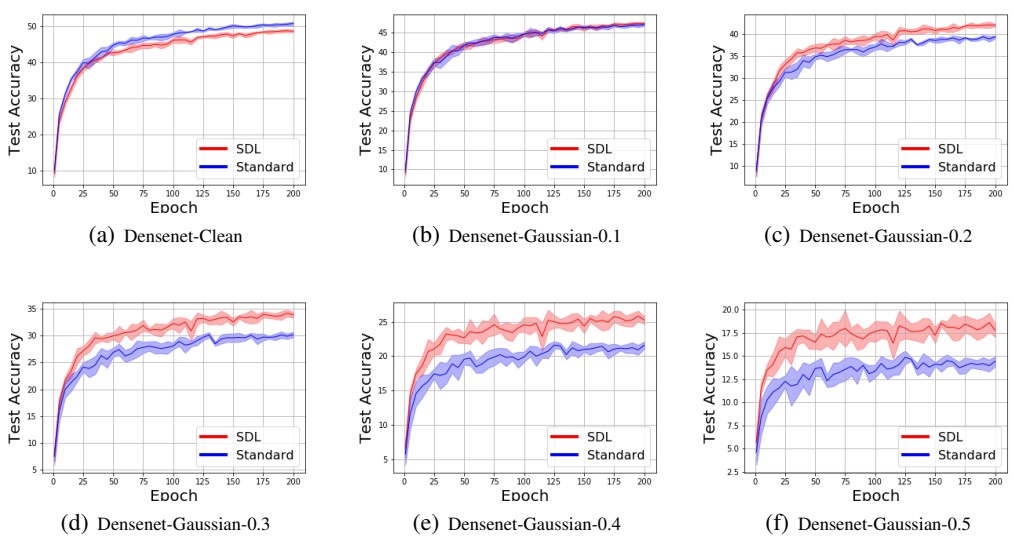

Figure 11:    Mean test accuracy $\pm$ std over 5 independent runs on Tiny-Imagenet dataset with Gaussian noise for Densenet with or without SDL

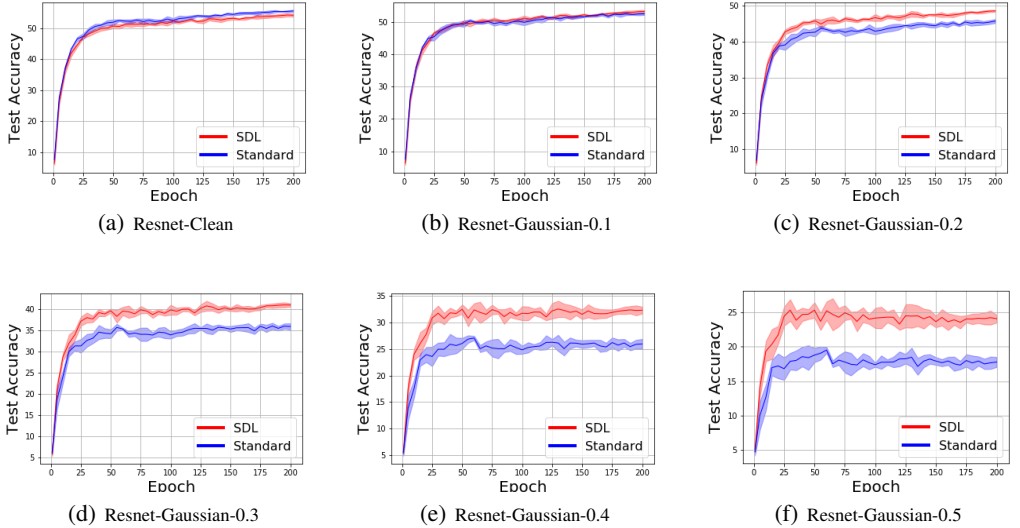

Figure 12:    Mean test accuracy $\pm$ std over 5 independent runs on Tiny-Imagenet dataset with Gaussian noise for Resnet with or without SDL

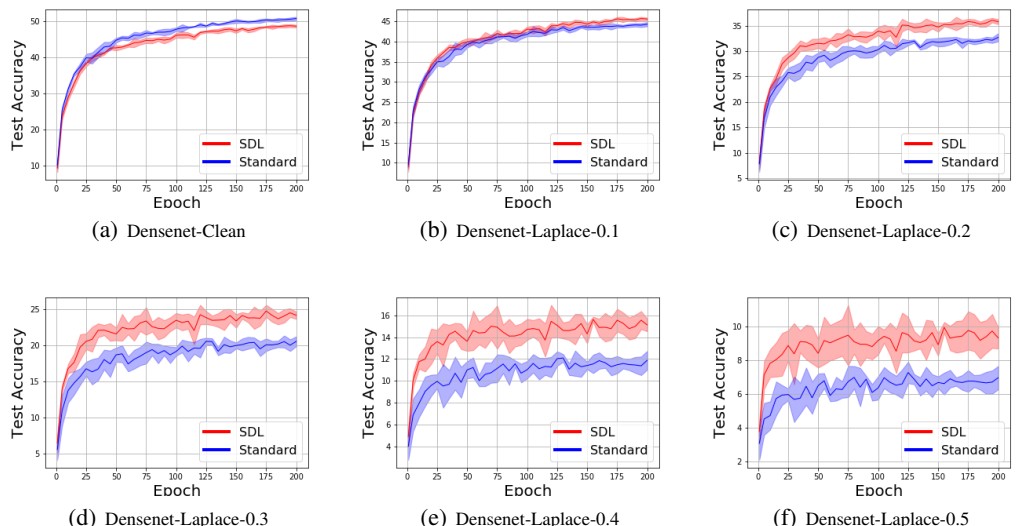

Figure 13: Mean test accuracy ± std over 5 independent runs on Tiny-Imagenet dataset with Laplace noise for Densenet with or without SDL

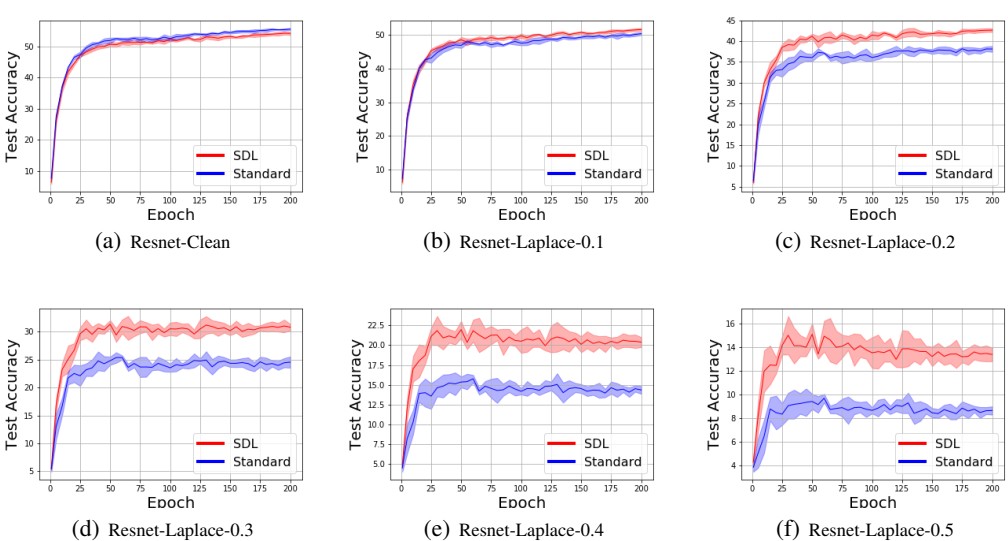

Figure 14: Mean test accuracy ± std over 5 independent runs on Tiny-Imagenet dataset with Laplace noise for Resnet with or without SDL

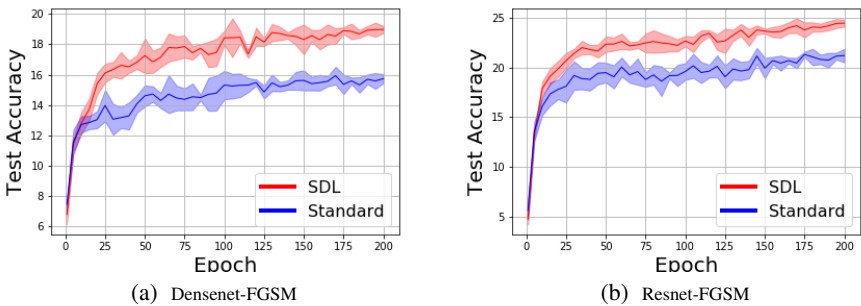

(a) Densenet-FGSM  (b) Resnet-FGSM

Figure 15: Mean test accuracy ± std over 5 independent runs on Tiny-Imagenet dataset under FGSM attack for Densenet and Resnet with or without SDL

## H    RESULTS OF RL ON SEAQUEST GAME

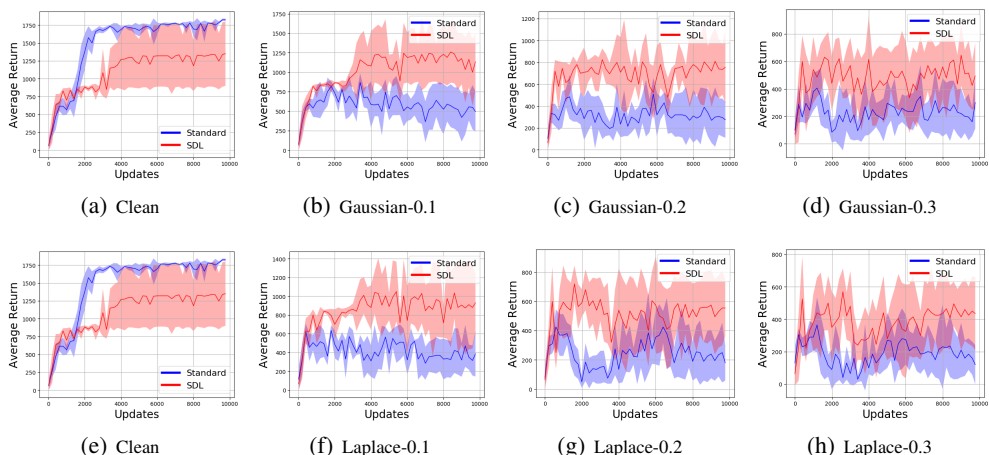

(a) Clean  (b) Gaussian-0.1  (c) Gaussian-0.2  (d) Gaussian-0.3

(e) Clean  (f) Laplace-0.1  (g) Laplace-0.2  (h) Laplace-0.3

Figure 16: Average Return ± std over 5 independent runs on *Seaquest* game with Gaussian/Laplace noise with or without SDL

# I  PYTORCH IMPLEMENTATION OF THE SDL LAYER

```python
class SparseDenoisingLayer(nn.Module):
    def __init__(self, sparseK, B,n):
        super(SparseDenoisingLayer, self).__init__()
        self.ksize = 2                      # kernelsize of Conv2d
        self.channel = 3                    # channel of input
        outplanes = self.channel*self.ksize*self.ksize
        self.B = torch.from_numpy(B).float().cuda()
        self.n = n

        self.outplanes = outplanes
        self.sparseK = sparseK

        self.register_parameter(name='U', param=torch.nn.Parameter(torch.
                                            randn(outplanes,outplanes).
                                            cuda() ) )

    def forward(self, x):
       # Cayley Mapping to compute orthogonal matrix R
        KA = torch.triu(self.U,diagonal=1 )
        tmpA = KA - KA.t()
        tmpB = torch.eye(self.outplanes,self.outplanes).cuda()-tmpA
        KU = torch.mm( (torch.eye(self.outplanes,self.outplanes).cuda()+
                                            tmpA ) , torch.inverse( tmpB
                                             ) )    # orthogonal matrix R

        weight = KU.view(self.outplanes,self.channel,self.ksize,self.
                                            ksize) #Reshape into the
                                            Conv2d parameter
        out = F.conv2d(x,weight, stride=1, padding = self.ksize-1)

        out = out.permute(0,2,3,1)
        out = torch.matmul(out,self.B)

        #  (function f) k-max pooling w.r.t the absolute value
        index = torch.abs(out).topk(self.sparseK, dim = 3)
        mask = torch.zeros(out.shape).cuda()
        mask.scatter_(3, index[1], 1.)
        out = out* mask

        # #
        out = torch.matmul(out,torch.transpose(self.B, 0, 1))
        out = out.permute(0,3,1,2)
        out = F.conv_transpose2d(out,weight, stride=1, padding = self.
                                            ksize-1  )/(self.ksize*self.
                                            ksize)

        return out  # reconstruction of the input
```

