# OpenReview forum: "A Simple Sparse Denoising Layer for Robust Deep Learning"
_ICLR.cc/2021/Conference — Reject_

### Official Review · AnonReviewer3 · 2020-10-27
**The idea of using a structured dictionary is interesting. However, the experiments are not convincing.**

**Rating:** 5
**Confidence:** 3

**Review:**

Thanks for the efforts of the authors. Some of my concerns have been addressed. However, I think that  the paper should test SDL layer by plugging in any hidden layer. So I keep my score.
_____________________________________________
Summary:
The paper provides a simple sparse denoising layer for robust deep learning. By employing a structured dictionary, the sparse coding phase of the l_0-norm based sparse coding has a closed-form solution. So it can be solved very fast when testing. Experiments show that the deep network with the presented sparse denoising layer outperforms the counterpart.

Pros:
The idea of using a structured dictionary is interesting, which produces a fast sparse coding. Also, the sparse coding layer can be integrated into a deep network and trained end-to-end.

Cons:
1.	The paper does not distinguish the corruption (e.g., additive Gaussian noise and blur) and the adversarial robustness. The paper considers the image corruption robustness. However, the statements in Section 1 and 2 are on adversarial robustness.
2.	The experiments do not test the superiority of the proposed approach. It only shows that using denoised images rather than the noisy ones are better for the classification task. It is especially when the denoising phase can be trained end-to-end with  deep networks. The paper only shows the effectiveness of SDL layer used as the first layer. It is better to verify the SDL layer used as a hidden layer. Moreover, I think the image denosing task [Zhang, 2017 in the references] is more appropriate for testing the superiority of SDL layer. Also, a two-stage approach should be included for comparison, i.e., it first denoises the noisy images with traditional sparse coding and then performs classification with deep networks without using SDL layer.
3.	Eq. (20) is the one-step sparse coding, i.e., it solves problem (7) or (15) for one-step update. What about mimicking the optimization of sparse coding, i.e., solves problem (7) or (15) for several updates?
4.	Figure 1 and Section 5 are not clear for me. The images in CIFAR10 are with a size of 32x32x3. However, sparse coding algorithms are image patch-based. Does the paper use 12x12x3 patches for classification task on CIFAR10? It is better to detail this issue.

Minor comments:
1.	The claim ”the alternative descent method heavily relies on SVD, which is not easy to extend to deep models” is not right. Actually, these are several papers shown that SVD can be extended to deep networks.
2.	Just above Eq. (8): ||Y_i|| -> ||Y_i||_0

---

> ### Author Response · Authors · 2020-11-24
> **Thanks for the suggestions of potential extension of SDL  and  off-line pre-process.**
>
>
> 1. Thanks for the suggestions of off-line pre-process by traditional sparse coding methods.  We discussed this point in the answers to common questions.
>
> 2. Moreover, thanks for the suggestion of stacking SDL. It is interesting to consider stacking several SDLs to mimic the optimization. Currently, we use a single SDL because of its low complexity and fast update.  We may consider stacking  SDL  as future work.
>
> 3. Implementation details.
> Our SDL does not need an explicit construction of a set of image patches for training.  It only uses the conv2d function. A Pytorch Implementation of our SDL is given in Appendix I.  Our codes have been submitted in the supplementary.   In our previous version, we set kernel size 12x12.  For faster computation, we further test our SDL with a small kernel size ( 2x2 ) that relates to the image patch (2x2x3). The updated experimental results are shown in the revised paper.   Our SDL with a small kernel size achieves a similar performance compared to the one with a large kernel size,  while it has fewer parameters and is faster.

---

### Official Review · AnonReviewer2 · 2020-10-28
**A Simple Sparse Denoising Layer for Robust Deep Learning**

**Rating:** 5
**Confidence:** 4

**Review:**

*Summary*
This paper addresses the problem of Deep Learning models' output instability to small perturbations in the input, first pointed out by Goodfellow et al 2015. They propose the usage of a denoising layer that can be applied as a first step to any deep CNN structure.

The proposed denoising layer consists in a sparse coding and dictionary learning algorithm:
-  whose sparse coding phase has a closed form solution (unlike the iterative methods in the literature). [both steps have independently closed form solutions, that need to be iterated]
- that allows end-to-end learning with low computational cost since it implements over-complete dictionary learning that translates easily as a deep learning layer.


In the numerical section they show how the method outperforms other state of the art methods when adding Gaussian noise to image classification and Reinforcement Learning


*Positive points*
- The paper tackles an important problem for ML with an interesting approach
- Numerical testing on very different scenarios such as classification tasks and reinforcement learning.
- The authors numerically show the robustness of their method to Gaussian noise perturbation of images
- The paper gives many details that allows reproducibility and is well written

*Negative points*
- The usage of denoising layers is not new in the context of image processing for restauration. It is clear that the paper would like to tackle the implications of denoising for image classification, but it is difficult to argue that Gaussian noise is representative of realistic perturbations that can trick an image classifier.

- The algorithms are tested on 'theoretical' data sets, meaning that the authors add synthetic noise to the CIFAR images and then train a classifier (idem for the RL tests). With no added noise the use of SDL does not improve the results (check Fig 2,3,4,5(a)) which might imply that the method cannot tackle real-life scenarios.

Some typos:
- sentence 'The optimization problem (1) is non-convex and non-smooth, which is very difficult to optimize' after eq 2 might need a bit more explaining or a reference since people have been using intensively numerical solvers to minimize non-convex and non-smooth energies.
- dimension of X is missing after eq 1 (although it is easy to understand from the dimension of the other vector/matrices)
- last sentence of the first page: ‘effeteness’, should be ‘effectiveness’?
- second page: ‘The both two steps are based on heuristic’, either 'both' or 'two'

*Conclusion*
The paper presents a new DL denoising layer that should allow to make classifiers and RL models more robust to added noise on the input. The instability of DL models to small perturbations is important and needs to be addressed, but the paper does not show that it can tackle realistic scenarios. Instead it focus on simulated perturbations of Gaussian noise on the CIFAR data set.

---

> ### Author Response · Authors · 2020-11-24
> **Thanks for suggestions of  evaluation of adversarial robustness and practical dataset.**
>
>
> Thanks for suggestions of evaluation of adversarial robustness.
>
> 1. We further test the adversarial robustness of our SDL under the fast gradient sign method (FGSM) attack. The experimental results on CIFAR10 and CIFAR100 are shown in Figure 2 in the revised paper.
>
> 2.  Furthermore,  we evaluate our SDL on a more practical dataset: tiny-Imagenet data. The experimental results under Gaussian noise, Laplace noise, and FGSM  attack are shown in Appendix G .  The experimental results also show that our SDL plug-in improves the robustness of the backbone models.

---

### Official Review · AnonReviewer1 · 2020-10-28
**Connection to dictionary learning seems not quite correct, denoising baseline would have been interesting**

**Rating:** 4
**Confidence:** 4

**Review:**

Pros: The general idea of the paper to introduce a simple single layer that does not really hurt the performance on clean data but is able to provide significantly more robust results on noisy data is interesting on valuable. Moreover, the numerical results demonstrate that classical networks are unable to use some of their expressive power to denoise and classify at the same time, such that the proposed approach indeed has significant benefits over the vanilla networks.

Cons: Despite the nice general idea, the paper has several weaknesses.
- Most significantly, I believe the derivation of the updates is not correct. Denoting $z = h-y$ for a moment, eq. (10) states that
$$ \|Bz\|^2 = \sum_j (z_j)^2 \|B_j\|^2. $$
However, we have
$$ \|Bz\|^2 = \langle \sum_j z_j B_j, \sum_i z_i B_i \rangle = \sum_{i,j} z_i z_j \langle B_i, B_j \rangle, $$
and the above equality holds if and only if $\langle B_i, B_j \rangle=0$ for $i\neq j$. The $B_i$, however, denote the columns of $B$ and the only property we know about $B$ is that $BB^T = I$, i.e., the rows of B are orthogonal. Therefore, I do not believe the result is correct. The same holds for the argument in eq. (18). Please correct me if I am overlooking something here.
- Secondly, the paper does not compare to any baseline other than training the plain networks (that were not really designed to handle significant noise). Is there absolutely no other approach in the literature to tackle this problem? And even if so, wouldn't applying a fixed denoiser as preprocessing be a better baseline?

Minor things:
- The motivation of the approach was not clear to me at the beginning. The introduction  refers to a missing robustness of neural networks and cites a paper on adversarial attacks. These attacks are, however, not addressed in the numerical experiments.
- It was a little unusual to me to avoid the problems of minimizing (1) by just designing parts of the matrix $D$. In which cases is the specific design with Fourier coefficients justified?
- Parts of the theory seem a little overstated to me - Theorem 1 follows from the orthogonality of the Fourier transform and Theorem 3 is a very simple corrolary. Maybe the term "Theorem" is a little too strong for these results. Theorem 2 is interesting but unfortunately seems far from the practical case.
- The final structure of the SDL frequently appears in other contexts where minimization methods are used to motivate network architectures or unrolling is used, e.g. in Schmidt and Roth, "Shrinkage Fields for Effective Image Restoration", or Kobler et al. "Variational Networks: Connecting Variational Methods and Deep Learning".

Currently, I clearly do not recommend the acceptance of this paper. If I am overlooking something in my first concern, and if a baseline comparison is added, I am happy to reconsider my score.

-----------
After the rebuttal: I would like to thank the authors for their response and clarification. The argument that the proposed strategy minimizes an upper bound is, however, somewhat weak, because $\|B(h - y)\|\leq \|B\| \|h-y\| $ is a possible upper bound for any matrix $B$, but using the right-hand side as a replacement for $\|B(y-y)\|$ within an optimization problem can make a large difference. I would expect to at least verify a marginal difference in the results numerically. Moreover, I do not think that the argument that some dictionary learning algorithms are slow is valid for omitting a comparison to a simple baseline denoiser at all: Your denoising baseline could be as simple as thresholding DCT coefficients or applying a median filter - operations that are surely as fast as the proposed approach (and still represent a sequential pipeline of first denoising and then applying a network, which one can expect to work worse than the proposed approach). Therefore, I keep my score and do not recommend this paper for acceptance in its current form.

---

> ### Author Response · Authors · 2020-11-24
> **Thanks for pointing out issues and constructive suggestions.**
>
>
> 1. Thanks for pointing out issues on Eq.(10).  Please see the response to the common questions.
>
> 2. In addition, thanks for the suggestions of the off-line pre-process.  We discussed this point in the response to common questions.

---

### Official Review · AnonReviewer4 · 2020-11-01
**The paper is generally well presented. However, a main issue is that the optimization algorithms for the l0-norm regularized problems (Section 3.1.2 and Section 3.2) are not correctly presented. Specifically, in the algorithm development to solve the ``Fix $\boldsymbol{R}$, optimize $\boldsymbol{Y}$" subproblem, it overlooks the coupling/interaction between the variables $y_1, y_2, \dots,y_M$ and mistakenly obtains a closed-form solution. See Comment 1 for details.**

**Rating:** 5
**Confidence:** 3

**Review:**

The paper is generally well presented. However, a main issue is that the optimization algorithms for the l0-norm regularized problems (Section 3.1.2 and Section 3.2) are not correctly presented. Specifically, in the algorithm development to solve the "Fix $\boldsymbol{R}$, optimize $\boldsymbol{Y}$" subproblem, it overlooks the coupling/interaction between the variables $y_1, y_2, \dots,y_M$ and mistakenly obtains a closed-form solution. See Comment 1 for details.

This paper proposes two L0 regularized sparse coding and dictionary learning models: one in bound form (with L0-norm explicitly bounded as a constraint in the formulated problem) and the other in Lagrangian form (with L0-norm term appearing in the objective function), and uses it as a robust layer for vanilla deep models. Experiments demonstrate that using the proposed model as a plug-in layer improves noise-robustness of both ResNet/DenseNet for classification and deep PPO models for reinforcement learning tasks.

Below are specific comments.

1. Main comments:

1.1 In Section 3.1.2, the "Fix $\boldsymbol{R}$, optimize $\boldsymbol{Y}$" subproblem is not correctly solved.

Specifically, in Eq. (10), the $\left\|\boldsymbol{B} (\boldsymbol{h}-\boldsymbol{y})\right\|_F^2$ (written as $\left\|\boldsymbol{B} (\boldsymbol{h}-\boldsymbol{y})\right\|_2^2$ below) should be
\[
\begin{aligned}
\left\|\boldsymbol{B} (\boldsymbol{h}-\boldsymbol{y})\right\|_2^2
&= \left\| \sum_{j=1}^M (h_j-y_j) \boldsymbol{B}_j \right\|_2^2 \\
&= \sum_{i=1}^M \sum_{j=1}^M (h_i-y_i) (h_j-y_j) \boldsymbol{B}_i^{\rm T} \boldsymbol{B}_j.
\end{aligned}
\]

Since $\boldsymbol{B}$ is a wide matrix (with much more columns than rows: $M=2n \gg d=2m$), its columns are not (all) orthogonal. Actually, as implied by Theorem 2, $|\boldsymbol{B}_i^{\rm T} \boldsymbol{B}_j|$ can be very close to $\|\boldsymbol{B}_j\|_2^2 = \|\boldsymbol{B}_j\|_2^2 = \frac{m}{n}$ when $n$ is comparable to or greater than $m^2$. Therefore, $\left\|\boldsymbol{B} (\boldsymbol{h}-\boldsymbol{y})\right\|_2^2$ can not be expressed as $\sum_{j=1}^M (h_j-y_j)^2 \|\boldsymbol{B}_j\|_2^2$, i.e., as a sum of $M$ functions, each depending on only $y_j$, and each $y_j$ appearing in only one of these functions, thereby each $y_j$ can be independently solved in closed form.

Instead, the variables $y_1, y_2, \dots,y_M$ are coupled in the objective and numerical optimization methods need to be used be solve them.

1.2 Similarly, in Section 3.2, the algorithm to solve the "Fix $\boldsymbol{R}$, optimize $\boldsymbol{Y}$" subproblem is also not correctly presented, due to the invalid equality in Eq. (18) which ignores the coupling/interaction between the variables $y_1, y_2, \dots,y_M$.


2. Minor comments (notation inconsistencies/abuse, typos, etc.):

In Section 3.1.1, "a $n \times n$ discrete Fourier matrix": The article `a' should be `an'.

"Eq.(4)" etc. is missing a space between "Eq." and equation number.

In Section 3.1.1 and Section 3.2, to make it clearer, please change the Frobenius norm operator $\|\cdot\|_F$ to the Euclidean norm operator $\|\cdot\|_2$ whenever it is applied to a vector, e.g., in Eq. (8) on the right-hand side of equality, and in Eqs. (9-11).

Throughout the paper, symbols denoting scalars are not in bold typeface, vectors are denoted as boldface lowercase letters and matrices as boldface capitals. For clarity and following the convention, these notation rules could also be applied to the lower-order parts, e.g., the $j$-th elements of vectors $\boldsymbol{h},\boldsymbol{y}$ are denoted as $h_j,y_j$ (rather than $\boldsymbol{h}_j,\boldsymbol{y}_j$), and the $i$-th columns of matrices $\boldsymbol{X},\boldsymbol{D},\boldsymbol{Y}$ are denoted as $\boldsymbol{x}_i,\boldsymbol{d}_i,\boldsymbol{y}_i$ (rather than $\boldsymbol{X}_i,\boldsymbol{D}_i,\boldsymbol{Y}_i$).

On page 5, the sentence in the first two lines should read as "We can see that Eq. (12) is minimized when $S$ consists of the indices of the $k$ largest (in absolute value) elements of $\boldsymbol{h}$".

In Eq. (18), $\|\boldsymbol{y}_j\|_0$ should be written as $1[y_j \neq 0]$, where $1[\cdot]$ is an indicator function which is 1 if its argument is true and 0 otherwise.

==================== After rebuttal =======================
Thank the authors for their detailed responses, which have addressed my concerns regarding Eqs. (10) and (18).

---

> ### Author Response · Authors · 2020-11-24
> **Thanks for pointing out the issues  and suggestion of the symbols.**
>
>
>
> 1.  Thanks for pointing out the issues in Eq.(10).  Please see the answers to the common questions.
>
> 2. In addition, thanks for the suggestion of the symbols. We have revised the paper using the suggested symbols.

---

### Author Response · Authors · 2020-11-24
**Response for common questions**


We thank all the reviewers for their valuable comments and constructive suggestions.

Common questions

1. Issues in Eq.(10).
When m=n, B is orthogonal. In this case, Eq.(10) holds true.
When, m<n, the R.H.S. of Eq.(10) can be seen as a closed-form approximation. We analyze the approximation error bound and revise the paper. We prove that the error bound is minimized by the same solution of the approximation.   It means that using the approximation update leads to a small approximation error.    We observe an empirical convergence of the iterative algorithm. (experimental results are given in Appendix D in the revised paper).

From the update rule, we design SDL as a lightweight plugin to improve the robustness of backbone models. when  m=n, SDL is a closed-form solution of sparse coding. When m<n, SDL is a simple approximation of the sparse coding.  Empirically, we found our SDL indeed improved the robustness.


2. Robustness of SDL.
The goal of this work is to investigate the robustness of the proposed lightweight plug-in architecture.  We have evaluated the robustness of our SDL under additive Gaussian and Laplace noises.   We further evaluate the adversarial robustness by testing the performances of our SDL under the fast gradient sign method (FGSM) attack. The perturbation parameter epsilon is set to 8/256. The test accuracy is reported in Figure 2 in the revised paper. It shows that our SDL plug-in improves the adversarial robustness of Resnet and Densenet.

Furthermore, we evaluate the robustness of SDL on the tiny-Imagenet dataset as suggested by the reviewer.  The experimental results are shown in Appendix G.  The performance of our SDL on the tiny-Imagenet dataset is consistent with that on the CIFAR10 and CIFAR100 datasets.
It shows that our SDL plug-in improves the robustness of backbone models under Gaussian noise, Laplace noise, and adversarial FGSM attack.


3. Advantages over a  two-stage approach with an off-line pre-processing step.
Our SDL is a lightweight plug-in that enables efficient batch gradient update.  It can process and perform gradient updates of a batch of images or states very fast. This is important for applications like RL and online learning.  For example,  in RL problems, the states depend on the policy network itself.  We cannot acquire all the states at the beginning for off-line processes.  In our experiments, the training process contains over 10 million gradient computations that require states as the input.  Traditional sparse coding methods often need to extract a set of image patches explicitly for training in an off-line manner.   Thus, these baselines are too slow to handle this problem.

The average time cost of SDL (forward+backward of SDL) on processing per-image

per-image in CIFAR10:  0.000287 (second)
per-image in CIFAR100: 0.000299 (second)
per-image in tiny-Imagenet: 0.000237(second)
Per-state in RL  :  0.001 (second).

To the best of our knowledge, we are the first lightweight plug-in architecture to address the robustness of deep networks.  Our SDL can be plugged into any successful architecture to improve robustness without additional human efforts on re-designing robust deep networks.

We want to highlight the difference between the goal of traditional sparse coding methods and our SDL.  The goal of traditional sparse coding methods is to achieve a high-resolution reconstruction for each single input image.  In contrast, our SDL  is a lightweight plug-in to address the robustness of deep networks.  Our SDL has much fewer parameters and a simpler structure, thus it enables fast update, which may have potential applications for low-resource edge devices and robots.

---

### Decision · Program_Chairs · 2021-01-07
**Final Decision**

**Decision:**

Reject

**Comment:**

The paper motivates the need for robustness, citing a paper on adversarial attacks. The type of perturbations are quite different (and of greater concern) than those originally included in the work, namely additive Gaussian or Laplace noise. This was raised by reviewers 1, 2 and 3.

The authors provided a detailed rebuttal in which they addressed many concerns of the reviewers. In particular, they included experiments with adversarial attacks. While these initial results seem interesting, the AC believes that they are too preliminary and it is not possible to evaluate their significance. The work should incorporate stronger baselines (other than plain networks) and consider more challenging forms of attacks (only FGSM attacks were studying). In particular, the paper should discuss where these ideas sit in the context of the literature and compare against conceptually related works that have been published in the area, such as

Xie, Cihang, et al. "Feature denoising for improving adversarial robustness." CVPR. 2019.

Without the adversarial perturbations, the impact of the work reduces significantly, as highlighted by reviewer 1, 2 and 3. In such case, it would be important to include stronger baselines in that setting as well. Please see the suggestions made by reviewer 1.

The authors satisfactorily addressed the problems in the derivation of the updates raised by Reviewers 1 and 4. However, new questions arise that would require careful consideration, as mention by reviewer 4 (which the authors could not answer as they were posted after the discussion period ended). This alone would not imply rejection, but suggests that the paper would be stronger after incorporating further feedback.

While this did not play a role in the decision, the AC suggests to also look at the literature around LISTA, in particular the work:
Liu, Jialin, and Xiaohan Chen. "ALISTA: Analytic weights are as good as learned weights in LISTA." ICLR. 2019.

All four reviewers recommend rejecting the paper and did not change their position after reading the author's response. The AC agrees with this assessment.